# Structure of SARS-CoV-2 M protein in lipid nanodiscs

Kimberly A Dolan[1,2], Mandira Dutta[3], David M Kern[2], Abhay Kotecha[4], Gregory A Voth[3], Stephen G Brohawn[2]*

[1]Biophysics Graduate Group, University of California, Berkeley, Berkeley, United States; [2]Department of Molecular and Cell Biology, Helen Wills Neuroscience Institute, and California Institute for Quantitative Biosciences (QB3), University of California, Berkeley, Berkeley, United States; [3]Department of Chemistry, Chicago Center for Theoretical Chemistry, Institute for Biophysical Dynamics, and James Franck Institute, The University of Chicago, Chicago, United States; [4]Materials and Structural Analysis Division, Thermo Fisher Scientific, Eindhoven, Netherlands

**Abstract** SARS-CoV-2 encodes four structural proteins incorporated into virions, spike (S), envelope (E), nucleocapsid (N), and membrane (M). M plays an essential role in viral assembly by organizing other structural proteins through physical interactions and directing them to sites of viral budding. As the most abundant protein in the viral envelope and a target of patient antibodies, M is a compelling target for vaccines and therapeutics. Still, the structure of M and molecular basis for its role in virion formation are unknown. Here, we present the cryo-EM structure of SARS-CoV-2 M in lipid nanodiscs to 3.5 Å resolution. M forms a 50 kDa homodimer that is structurally related to the SARS-CoV-2 ORF3a viroporin, suggesting a shared ancestral origin. Structural comparisons reveal how intersubunit gaps create a small, enclosed pocket in M and large open cavity in ORF3a, consistent with a structural role and ion channel activity, respectively. M displays a strikingly electropositive cytosolic surface that may be important for interactions with N, S, and viral RNA. Molecular dynamics simulations show a high degree of structural rigidity in a simple lipid bilayer and support a role for M homodimers in scaffolding viral assembly. Together, these results provide insight into roles for M in coronavirus assembly and structure.

*For correspondence: brohawn@berkeley.edu

## Editor's evaluation

The SARS-CoV2 M protein is an abundant protein in the viral envelope and is a potential target for vaccine and therapeutic development. M is one of only four structural proteins that are incorporated into mature SARS-CoV2 virions. This paper describes a single-particle cryo-electron microscopy structure of M in lipid nanodiscs. M forms a dimer similar to the previously characterized SARS-CoV2 ORF3a viroporin, which was proposed to function as an ion channel. Structural analysis and molecular dynamics simulations indicate that, unlike ORF3a, M functions as a structural scaffold protein. The highly charged surface of the M's cytosolic domain suggests how it might interact with other structural proteins and/or the charged RNA genome during virion packaging.

## Introduction

Coronaviruses encode four structural proteins that are incorporated into mature enveloped virions: the transmembrane spike (S), membrane (M), and envelope (E) proteins, and the soluble nucleocapsid (N) protein (*Masters, 2006*). S proteins protrude from the virion, creating the eponymous corona in electron micrographs, and mediate fusion of viral and host cell membranes. E proteins form cationic

viroporins that promote viral assembly and modulate the host immune response. N is an RNA-binding protein that packages the viral RNA genome. M organizes the assembly and structure of new virions and is essential for virus formation (*Sturman et al., 1980*; *Armstrong et al., 1984*; *Yu et al., 2021*; *Finkel et al., 2021*). M is the most abundant membrane protein in the viral envelope and anti-M antibodies are found in plasma of patients infected with SARS-CoV-2 and other coronaviruses (*Godet et al., 1992*; *He et al., 2005*; *Hotop et al., 2022*; *Heffron et al., 2021*; *Martin et al., 2021*; *Jörrißen et al., 2021*). Based on its functional importance and immunogenicity, M has been proposed as a target for coronavirus vaccines or therapeutics.

In infected cells, M mediates virus assembly and budding by interacting with all other structural proteins and directing their localization to the ER-Golgi intermediate compartment (*Masters, 2006*; *de Haan et al., 1999*; *Cavanagh, 2005*; *Siu et al., 2008*; *Neuman et al., 2011*; *Kuo et al., 2016a*; *Kuo et al., 2016b*; *Lim and Liu, 2001*; *Boson et al., 2021*). M is proposed to interact with E through its transmembrane region and S and N through a cytosolic C-terminal region (*Kuo et al., 2016a*; *Kuo et al., 2016b*; *Lim and Liu, 2001*; *Boson et al., 2021*). ER export and Golgi localization sequences in M determine its subcellular localization and M, in turn, modulates localization and posttranslational processing of S to promote virion assembly (*Boson et al., 2021*; *Perrier et al., 2019*). Across a wide range of coronaviruses (including SARS-CoV-2, SARS-CoV-1, MERS, mouse hepatitis virus [MHV], infectious bronchitis virus, and transmittable gastroenteritis virus), M is required for minimal virus-like particle (VLP) formation in transfected cells (*Siu et al., 2008*; *Vennema et al., 1996*; *Hsieh et al., 2005*; *Xu et al., 2020*; *Plescia et al., 2021*). M is insufficient for VLP formation alone, however, and co-required components vary in different systems. SARS-CoV-2 VLP formation requires M co-expression with S or N (*Xu et al., 2020*; *Plescia et al., 2021*).

M has further been implicated in modulating host antiviral innate immunity. M inhibits the innate immune response by interfering with MAVS-mediated signaling and interferon production (*Fu et al., 2021*; *Sui et al., 2021*). In mouse models of infection, M expression results in lung epithelial cell apoptosis in vitro and in vivo and may contribute to lung injury and pulmonary edema found in severe disease (*Sui et al., 2021*).

Despite its essential role in viral assembly and implication in pathogenesis, the molecular determinants of M function remain largely unknown. MHV M was proposed to adopt long and compact structures that differentially facilitate membrane bending and recruitment of other structural proteins based on low-resolution tomographic analysis (*Neuman et al., 2011*). Intriguingly, a structural and

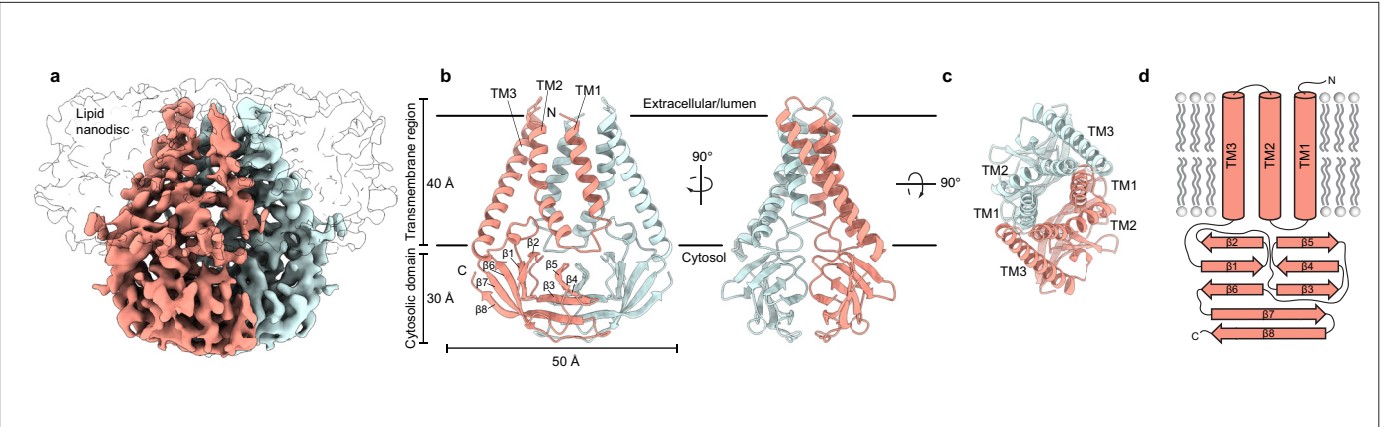

**Figure 1.** Structure of SARS-CoV-2 membrane (M) in lipid nanodiscs. (**a**) A 3.5 Å resolution cryo-EM map of SARS-CoV-2 M in MSP1E3D1 nanodiscs viewed from the membrane. One subunit is colored pink, and the second subunit is colored blue. Density corresponding to the lipid nanodisc is shown transparent. (**b,c**) Model of M viewed (**b**) from the membrane in two rotations and (**c**) from the extracellular or lumenal side. (**d**) Cartoon schematic of an M monomer with secondary structure elements indicated.

The online version of this article includes the following figure supplement(s) for figure 1:

**Figure supplement 1.** Purification and reconstitution of membrane (M).

**Figure supplement 2.** Cryo-EM processing and validation.

**Figure supplement 3.** Comparison of experimentally determined and predicted membrane (M) protein structures.

**Figure supplement 4.** Sequence alignment of membrane (M) proteins across Coronaviridae.

evolutionary relationship between SARS-CoV-2 M and the accessory viroporin ORF3a was reported (*Tan et al., 2021*; *Pezeshkian et al., 2021*) based on predicted homology to our experimental ORF3a structures (*Kern et al., 2021*). The manner in which distinct functional roles for M and ORF3a can be achieved in the context of a shared architecture remains to be determined. Here, we report the cryo-EM structure of SARS-CoV-2 M in lipid nanodiscs and perform molecular dynamics (MD) simulations to provide insight into M structure, function, and dynamics.

## Results

We determined the structure of SARS-CoV-2 M in lipid nanodiscs (*Figure 1*). Full-length M was expressed in *Spodoptera frugiperda* (Sf9) cells with a cleavable C-terminal GFP tag. Gel filtration chromatography of protein extracted in DDM/CHS detergent shows M runs predominantly as a single species consistent with a 50 kDa homodimer. We do not observe evidence of specific higher-order oligomerization at low concentrations by fluorescence size exclusion chromatography or at higher concentrations in large-scale purifications (*Figure 1—figure supplement 1*). SARS-CoV-2 ORF3a, in contrast, assembles into stable homodimers and homotetramers under similar conditions (*Kern et al., 2021*).

We reconstituted homodimeric SARS-CoV-2 M in nanodiscs made from the scaffold protein MSP1E3D1 and lipids (DOPE:POPC:POPS in a 2:1:1 ratio) and determined its structure by cryo-EM (*Figure 1*, *Figure 1—figure supplement 2*, *Supplementary file 1*). The majority of M (189 of 222 amino acids per subunit) was de novo modeled in the cryo-EM map (*Figure 1*, *Figure 1—figure supplement 2*). The N-terminus (amino acids 1–16) and C-terminus (amino acids 205–222) are not resolved in the map and were not modeled. Loops connecting transmembrane helices (amino acids 36–42 and 71–78) are the least well-resolved regions of the structure. The relatively weak density is consistent with a lack of stabilizing interactions between these and other M regions and likely indicates they adopt a range of conformations among particles used to generate the final map.

M is ~70 Å tall when viewed from the membrane with an ~40 Å transmembrane spanning region and ~30 Å cytosolic domain (CD) extending into the intracellular solution (*Figure 1*). Each subunit contains an extracellular or lumenal N-terminus, three transmembrane helices (amino acids 17–36, 43–71, and 79–105) connected by short linkers, and a β-strand rich C-terminal cytosolic domain. We note that AlphaFold and RoseTTAFold predicted M structures diverge substantially from the experimental structure (*Figure 1—figure supplement 3*; *Heo and Feig, 2020*; *Jumper et al., 2021*). In the predicted structures, TM1s are swapped between subunits in addition to differences in the relative positions of transmembrane and cytosolic domains.

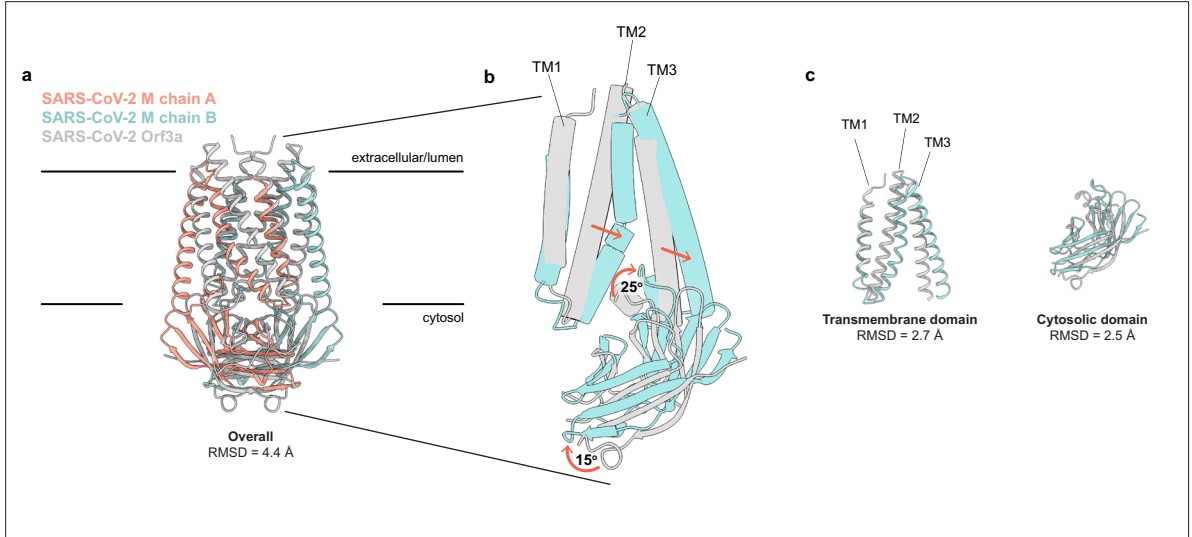

**Figure 2.** SARS-CoV-2 membrane (M) and ORF3a proteins are structurally homologous. (**a**) Overlay of M and ORF3a structures. M is colored with one subunit pink and the second subunit blue and ORF3a is white. (**b**) Overlay of a single subunit indicating major conformational rearrangements. (**c**) Overlay of isolated transmembrane and cytosolic domains from each protein.

Viewed from above, TM1-TM3 from each subunit is positioned along a flattened ellipse with a long major (~50 Å) and short minor (~16 Å) axis. Within each subunit, TM2-TM3 are closely juxtaposed and tightly packed while TM1-TM2 are more distant and loosely connected. The two subunits assemble along their long axis, with TM1 from one protomer forming extensive interactions with the TM2-TM3 unit of the second protomer. As they project toward the cytoplasm, the three transmembrane helices twist counterclockwise and splay outward in the inner leaflet, creating an expanded ellipse with ~55 and ~40 Å axes at the intracellular leaflet.

The transmembrane region is connected to the cytosolic domain through a tight turn-helix-turn segment comprised of residues 106–116. Within the cytosolic domain, each protomer chain forms a pair of opposing β-sheets packed against one another in an eight stranded β-sandwich (*Figure 1B and D*). The outer sheet is formed by strands β1, β2, β6, the N-terminal half of β7, and the C-terminal half of β8. The inner sheet is formed by strands β3, β4, β5, the C-terminal half of β7, and the N-terminal half of β8. The inner sheets from each protomer interact through a large (~690 Å$^2$ buried surface area per chain) and complementary interface with residues L138, V139, V143, L145, F193, A195 contributing to a hydrophobic core surrounded by additional polar interactions.

Using Dali (*Holm, 2020*) to compare the M structure to all experimentally determined protein structures returns SARS-CoV-2 ORF3a (*Kern et al., 2021*) as the only structural homolog with a shared fold. Superposition of the two viral proteins reveals a similar fold topology and homodimeric assembly with an overall RMSD of 4.4 Å. Isolated transmembrane and cytosolic domains from individual protomers are better superimposed (RMSD = 2.7 and 2.5 Å, respectively) (*Figure 2*). Substantial differences in M and ORF3a structure are observed in three regions: TM2-TM3, the transmembrane-cytosolic domain junction, and the cytoplasmic domain interface. TM1s of M and ORF3a are well superimposed, but TM2-TM3 of M are splayed further out into the membrane and are less twisted about the twofold symmetry axis to create a flatter and tighter interaction surface. The angle between TM3 and the

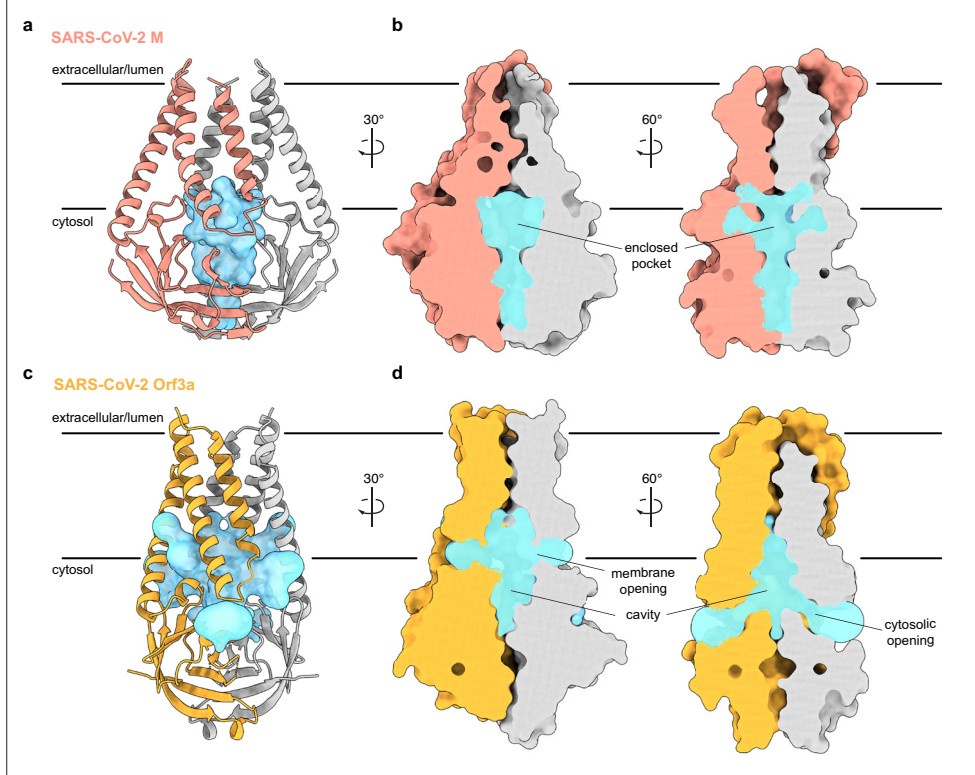

**Figure 3.** An enclosed polar pocket between cytosolic domains in membrane (M). (**a**) M shown as a cartoon and (**b**) surface with enclosed pocket volume calculated with CASTp36 shown as a blue surface. The enclosed pocket in M is formed between cytosolic domains and is sealed to the surrounding solution by protein. (**c,d**) Same as (**a,b**), but for SARS-CoV-2 ORF3a. The cavity in ORF3a begins closer to the lipid bilayer, extends approximately halfway across the membrane, and is open to surrounding solution and lipids through multiple openings (*Kern et al., 2021*).

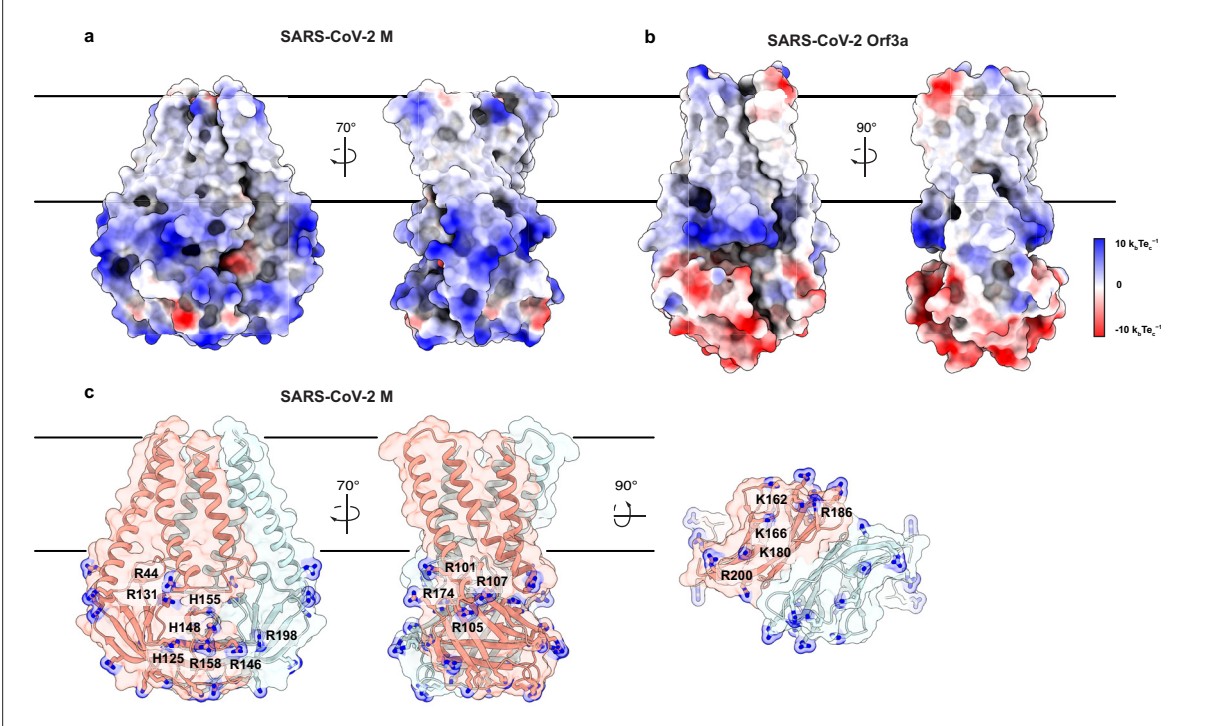

**Figure 4.** An electropositive cytosolic surface in membrane (M). (**a,b**) Views of the wide and narrow faces of (**a**) M and (**b**) ORF3a colored according to electrostatic surface potential from red (electronegative, −10 kbTec-1) to blue (electropositive, +10 kbTec-1). (**c**) Views of three electropositive surface patches on M cytosolic domains with basic residues labeled and shown as sticks with blue nitrogen atoms.

cytosolic domain is ~25° more acute in M. The cytosolic domains of M are rotated ~15° away from the symmetry axis relative to ORF3a, shifting the cytosolic domain interface between subunits further from the membrane.

What are the consequences of the structural rearrangement in M relative to ORF3a? Association of subunits in the M homodimer creates a polar and, presumably, water-filled pocket, reminiscent of the polar cavity created between subunits in ORF3a. However, the included volumes are different in several key respects (*Figure 3*). First, the M pocket is ~⅓ smaller with an enclosed volume of ~840 Å$^3$ compared to ~1300 Å$^3$ in ORF3a. Second, the M pocket is completely sealed by protein to the surrounding membrane and cytoplasm; no openings large enough for water passage are observed connecting the pocket and protein exterior. In contrast, ORF3a displays three pairs of tunnels connecting its internal cavity to the membrane and cytoplasm (two are displayed in *Figure 3*). Third, the position of the M pocket and ORF3a cavity are different. In M, the gap between subunits is confined to the region between cytosolic domains because transmembrane helices from opposing subunits form tight interactions across the entire lipid bilayer. In ORF3a, the gap extends from the region between cytosolic domains to approximately halfway across the membrane because transmembrane helices are less tightly associated across the membrane inner leaflet.

Another major difference in M and ORf3a structures is shown in *Figure 4*. The cytosolic domain of M is strikingly electropositive across nearly the entire exposed surface. Electropositive character is contributed by 17 basic amino acids in three surface patches. The first covers the wide face of the cytosolic domains and consists of eight residues (R44, H125, R131, R146, H148, H155, R158, and R198). The second covers the narrow face of M and consists of four residues (R101, R105, R107, and R174). The third covers the underside of M and consists of five residues (K162, K166, K180, R186, and R200). ORF3a, in contrast, presents mixed electrostatic character with electropositive patches closer to the membrane and electronegative patches toward the cytoplasm. Such uniform electropositivity across the M cytosolic surface could facilitate the close juxtaposition of M present at high concentration in viral envelope with the negatively charged viral RNA genome.

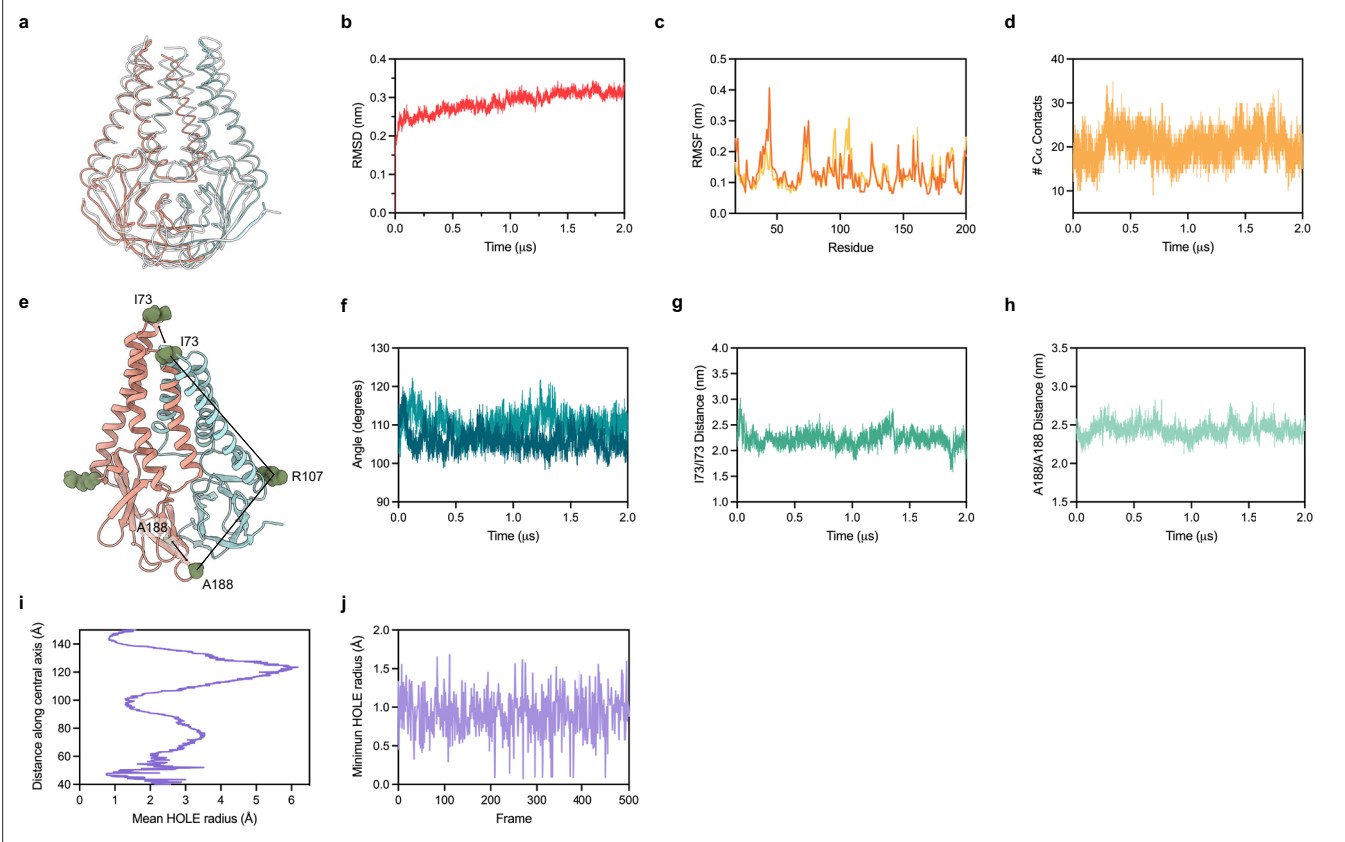

**Figure 5.** Molecular dynamics simulation of membrane (M). (**a**) Overlay of M cryo-EM structure (colored in pink and blue) and final structure (in white) following 2.0 µs all-atom molecular dynamics simulation. (**b**) Overall RMSD between simulated and initial structure during simulation. (**c**) Root mean square fluctuation of protein residues in the simulation. Orange and yellow colors correspond to individual M protein chains. (**d**) Number of C-alpha contacts between two monomers. (**e**) Structural representation of distances and angles used for calculations in (**f–h**). (**f**) A188-R107-I73 angle plot for each monomer. One monomer has slightly higher values than the other. (**g**) Center of mass distance between I73 residues at the top of the TM2-TM3 linker. (**h**) Center of mass distance between A188 residues at the base of the cytosolic domains. (**i**) Mean radius of the enclosed pocket in M over the simulation trajectory versus distance along the symmetry axis. At its widest positions, the pocket is wide enough to accommodate two water molecules. (**j**) Minimum hole radius versus the frame number in the simulation. The lack of substantial changes in radius indicates a stable pocket size and shape that does not open to solution during the simulation.

The online version of this article includes the following figure supplement(s) for figure 5:

**Figure supplement 1.** Analysis of pocket size and lipid interactions from molecular dynamics simulation.

The large, complementary, and hydrophobic interface between transmembrane and cytosolic regions of subunits in the M structure suggests a structurally rigid core. However, a previous tomographic study of MHV suggesting that M adopts distinct long and compact structures (*Neuman et al., 2011*), M's structural homology to the viroporin ORF3a (*Tan et al., 2021*; *Kern et al., 2021*), and the dissociation of cytosolic regions shown in predicted SARS-CoV-2 M structures (*Heo and Feig, 2020*) suggest the possibility that M is capable of undergoing large-scale structural rearrangements. We next performed MD simulations to explore potential conformational dynamics of M.

We equilibrated M in a lipid environment and ran an all-atom MD simulation for 2 µs. Major conformational changes related to protein function typically take place on a timescale ranging from a microsecond to seconds (*Xue et al., 2012*; *Klepeis et al., 2009*; *Don et al., 2018*). Prior work has reported that microsecond simulations can capture large conformational changes of membrane proteins in a lipid bilayer (*Monje-Galvan and Voth, 2021*; *Brandner et al., 2019*; *Nury et al., 2010*; *Monje-Galvan and Voth, 2020*). We therefore expected to capture a subset of possible structural changes in our simulation that correspond to relatively frequently accessed states. Overall, we did not observe substantial conformational rearrangements in M during the simulation (*Figure 5A and B*). Superposition of the experimental and final M structure following the simulation shows minor deviations through

most of the protein (overall RMSD of 2.5 Å) (*Figure 5A*). The largest difference is a shift in TM1 up toward the extracellular/lumenal side by approximately half a helical turn, enabled by rearrangement of the TM1-TM2 linker (*Figure 5A*). This relatively subtle movement is consistent with weaker density for the TM1-TM2 linker in the cryo-EM map and fewer packing interactions for TM1 than TM2 or TM3. Per residue deviations ranged from ~1 to 4 Å and, aside from the movement of TM1, were similar between subunits and largest in the TM2-TM3 linker, transmembrane to cytosolic region connection, and loops connecting strands in the cytosolic domain. Minimal structural deviation was observed during the simulation within or between subunits as judged by the number of close Cα contacts, the angle between transmembrane and cytosolic regions, the distance between transmembrane regions, or the distance between cytosolic domains (*Figure 5D–H*). Consistent with limited movement of the transmembrane region and a lack of evidence for lipid binding in the cryo-EM structure, no obvious enrichment of specific lipids around M was identified following simulation (*Figure 5—figure supplement 1*). Finally, the internal M pocket remained similar in size and sealed from the surrounding solution throughout the simulation (*Figure 5I and J*). Taken together, these data suggest that M adopts a stable structure with minimal dynamic rearrangements on the microsecond timescale, at physiological temperature, and in the absence of additional binding proteins. These results are consistent with the absence of other highly populated conformations in our cryo-EM data, though we cannot exclude the possibility that M undergoes rare large-scale conformational changes under these conditions not captured in the simulation.

## Discussion

The structure of the SARS-CoV-2 M protein that we have obtained by cryo-EM reveals a homodimeric fold that is structurally homologous to the nonselective $Ca^{2+}$ permeable cation channel of SARS-CoV-2, ORF3a. As with 3a, each subunit of M contains three transmembrane helices and a C-terminal β-sandwich domain. However, the structure differs from ORF3a in several key ways that provide insight into how these structurally similar proteins can fill drastically different apparent roles in the coronavirus life cycle.

When viewed from the plane of the membrane, M is considerably wider and flatter than ORF3a, due to differences in transmembrane helix packing and a rotation about the central axis of the cytosolic domain. Among the consequences of this flattening out of M are distinct differences in the dimer interface across the membrane, where M shows a tighter dimer interface closer to the membrane outer leaflet as well as a gap between cytosolic domain subunits that forms an enclosed pocket lined by polar residues. In ORF3a, transmembrane regions are less closely opposed and a gap between subunits extends from halfway across the membrane to halfway down the cytosolic domains. The result is a larger cavity that is open to the membrane and cytoplasm. Mutations in the ORF3a cavity alter ion channel activity, consistent with the cavity forming part of the conduction path. Tight subunit association may therefore be important for the structural role of M, while loose subunit association that creates a large open cavity may be essential for the viroporin activity of ORF3a.

In further contrast to ORF3a, which was seen to form stable tetramers through electrostatic interactions between neighboring dimers, we see no evidence that M forms higher-order oligomers under similar experimental conditions. Surface characteristics of the M dimer lend credence to the possibility that M exists solely as a dimer in the membrane – one striking feature of the M C-terminal β-sandwich domain is the presence of three sizable patches of positive charge that dominate its solvent exposed surface. MD simulations of M show that the dimer is stable and does not readily adopt alternate conformations at physiological temperature over the 1.6 μs trajectory. These data suggest that M could play a structural role in mediating morphological changes in host cell membranes through interactions with other SARS-CoV-2 structural proteins and perhaps negatively charged lipid headgroups or viral RNA.

A complementary study published during review of this work includes two structures of SARS-CoV-2 M solubilized in detergent micelles and in complex with Fab antibody fragments bound to the cytosolic domain (*Zhang et al., 2022*). One structure (termed short form) is similar to that reported here. The second structure (termed long form) adopts a distinct conformation that is taller and narrower in the membrane. The conformations are suggestive of short and long forms of MHV M observed in low-resolution tomograms (*Neuman et al., 2011*). Our data suggests that M in a simple lipid mixture predominantly adopts the short form. It may be that the alternate M long form, stabilized

by Fab binding for structure determination, is similarly promoted in a physiological context by interactions with lipids, spike, N, or viral RNA. Evidence for possible higher-order oligomerization was also reported in some 2D classes of M solubilized in a different detergent, suggesting that M-M interactions may occur under certain biochemical conditions. Whether distinct M conformations have distinct functional roles in virus assembly, as has been suggested for MHV M, remains to be determined.

M has been shown to play a crucial role in viral assembly through protein-protein interactions with other coronavirus structural proteins such as N and S. Spike proteins are incorporated into coronavirus virions via interactions between the cytosolic tail of S and the cytosolic domain of M, however the precise details of this interaction are unknown (*Boson et al., 2021*). In SARS-CoV-2, M and N or S are the minimal components required for forming VLPs when expressed heterologously in cells (*Xu et al., 2020*; *Plescia et al., 2021*). Several recent studies have suggested that the C-terminal domain of N is the site of interaction between SARS-CoV-2 M and N, but as with S a precise binding site has not been established (*Cubuk et al., 2021*; *Lu et al., 2021*). It is possible that M-N interactions are mediated by favorable electrostatic interactions between negatively charged residues of the N CTD and one or more of the basic patches identified on the surface of the cytosolic domain of M, a model supported by co-immunoprecipitation of N and M proteins expressed in 293T cells (*Zhang et al., 2022*). Through the sheer abundance of M dimers found in the membrane of SARS-CoV-2 virions, M and N together might facilitate VLP formation via a mechanism similar to the Gag precursor of HIV, where the high concentration of M C-terminal domains at the cytoplasmic membrane surface recruit and organize many N proteins that together physically extrude a membranous bud.

At present the World Health Organization puts the confirmed number of COVID-19 cases worldwide at nearly 530 million. Over the last 2 years the SARS-CoV-2 virus has undergone many mutations that have been extensively documented through sequencing efforts worldwide (*Hadfield et al., 2018*). Despite this, the M protein sequence has remained virtually unchanged – a testament to the critical role that M plays in viral replication and assembly (*Cagliani et al., 2020*). Furthermore, while only 20 amino acids in length, the N-terminus of M has been found to be highly immunogenic in COVID-19 patients (*Hotop et al., 2022*; *Heffron et al., 2021*; *Martin et al., 2021*; *Jörrißen et al., 2021*). M has also been shown to modulate innate immune response and could contribute to lung injury often seen in severe cases (*Fu et al., 2021*; *Sui et al., 2021*). Given its clear importance in the coronavirus life cycle and pathogenicity, M presents an attractive target for therapeutics or vaccines. While M is well conserved across Coronaviridae (*Figure 1—figure supplement 4*), it shows particularly high conservation between SARS-CoV-1 and SARS-CoV-2, with a sequence similarity of 90.54%, highlighting its potential as a therapeutic target for emergent coronaviruses in the future.

## Methods
### Cloning and protein expression
The coding sequence for SARS-Cov-2 M protein (Uniprot P0DTC5) was synthesized (IDT, Newark, NJ) and cloned into a vector based on the pACEBAC1 backbone (MultiBac; Geneva Biotech, Geneva, Switzerland) with an added C-terminal PreScission protease (PPX) cleavage site, linker sequence, superfolder GFP (sfGFP), and 7xHis tag, generating a construct for expression of M-SNS-LEVLFQGP-SRGGSGAAAGSGSGS-sfGFP-GSS-7xHis (*Kern and Brohawn, 2021*). MultiBac cells were used to generate a Bacmid according to the manufacturer's instructions. Sf9 cells were cultured in ESF 921 medium (Expression Systems, Davis, CA) and P1 virus was generated from cells transfected with Escort IV reagent (MilliporeSigma, Burlington, MA) according to the manufacturer's instructions. P2 virus was then generated by infecting cells at 2 million cells/mL with P1 virus at an MOI ~ 0.1, with infection monitored by fluorescence and harvested at 72 hr. P3 virus was generated in a similar manner to expand the viral stock. The P2 or P3 viral stock was then used to infect Sf9 cells at 4 million cells/mL at an MOI ~ 2–5. At 72 hr, infected cells containing expressed M-sfGFP protein were harvested by centrifugation at 2500 × *g* for 10 min and frozen at –80°C.

### Protein purification
Infected Sf9 cells from 1 L of culture (~15 mL of cell pellet) were thawed in 100 mL of Lysis Buffer containing 50 mM HEPES, 150 mM KCl, 1 mM EDTA pH 8. Protease inhibitors (final concentrations: E64 [1 μM], pepstatin A [1 μg/mL], soy trypsin inhibitor [10 μg/mL], benzamidine [1 mM], aprotinin

[1 µg/mL], leupeptin [1 µg/mL], AEBSF [1 mM], and PMSF [1 mM]) were added to the lysis buffer immediately before use. Benzonase (4 µL) was added after the cell pellet thawed. Cells were then lysed by sonication and centrifuged at 150,000 × $g$ for 45 min. The supernatant was discarded, and residual nucleic acid was removed from the top of the membrane pellet using DPBS. Membrane pellets were scooped into a dounce homogenizer containing extraction buffer (50 mM HEPES, 150 mM KCl, 1 mM EDTA, 1% $n$-dodecyl-β-D-maltopyranoside (DDM, Anatrace, Maumee, OH), 0.2% cholesteryl hemisuccinate Tris salt (CHS, Anatrace, Maumee, OH) pH 8). A stock solution of 10% DDM, 2% CHS was dissolved and clarified by bath sonication in 200 mM HEPES pH 8 prior to addition to buffer to the indicated final concentration. Membrane pellets were then homogenized in extraction buffer and this mixture (150 mL final volume) was gently stirred at 4°C for 1.5 hr. The extraction mixture was centrifuged at 33,000 × $g$ for 45 min and the supernatant, containing solubilized membrane protein, was bound to 4 mL of Sepharose resin coupled to anti-GFP nanobody for 1.5 hr at 4°C. The resin was then collected in a column and washed with 10 mL of buffer 1 (20 mM HEPES, 150 mM KCl, 1 mM EDTA, 0.025% DDM, 0.005% CHS, pH 7.4), 40 mL of buffer 2 (20 mM HEPES, 500 mM KCl, 1 mM EDTA, 0.025% DDM, 0.005% CHS, pH 7.4), and 10 mL of buffer 1. The resin was then resuspended in 6 mL of buffer 1 with 0.5 mg of PPX protease and rocked gently in the capped column for 2 hr. Timing of this step was critical as longer incubations in detergent significantly reduced yield and sample quality. Cleaved M protein was then eluted with an additional 12 mL of wash buffer, spin concentrated to ~1 mL with Amicon Ultra spin concentrator 10 kDa cutoff (Millipore), and loaded onto a Superose 6 increase column (GE Healthcare, Chicago, IL) on an NGC system (Bio-Rad, Hercules, CA) equilibrated in buffer 1. Peak fractions containing M protein were then collected and spin concentrated prior to incorporation into nanodiscs.

## Nanodisc formation

Freshly purified M protein in buffer 1 was reconstituted into MSP1E3D1 nanodiscs with a mixture of lipids (DOPE:POPS:POPC at a 2:1:1 mass ratio, Avanti, Alabaster, AL) at a final molar ratio of 1:4:400 (M:MSP1E3D1:lipid).

Twenty mM solubilized lipid in lipid dilution buffer (20 mM HEPES, 150 mM KCl, pH 7.4) was mixed with additional DDM/CHS detergent and M protein at 4°C for 30 min before addition of purified MSP1E3D1. This addition brought the final concentrations to approximately 10 µM M protein, 40 µM MSP1E3D1, 4 mM lipid mix, 10 mM DDM, and 1.7 mM CHS. The solution with MSP1E3D1 was mixed at 4°C for 15 min before addition of 150 mg of Biobeads SM2. Biobeads (washed into methanol, water, and then Nanodisc Formation Buffer) were weighed after liquid was removed by pipetting (damp weight). This final mixture was then gently tumbled at 4°C overnight (~12 hr). Supernatant was cleared of beads by letting large beads settle and carefully removing liquid with a pipette. Sample was spun for 10 min at 21,000 × $g$ before loading onto a Superose 6 increase column in 20 mM HEPES, 150 mM KCl, pH 7.4. Peak fractions corresponding to M protein in MSP1E3D1 were collected, 10 kDa cutoff spin concentrated, and used for grid preparation. MSP1E3D1 was prepared as previously described (*Ritchie et al., 2009*) without cleavage of the His-tag.

## Cryo-EM sample preparation and data collection

M in MSP1E3D1 was prepared at a final concentration of 1.3 mg/mL. Concentrated sample was cleared by a 10 min 21,000 × $g$ spin at 4°C prior to grid preparation; 3.4 µl of protein was applied to freshly glow discharged Holey Carbon, 300 mesh R 1.2/1.3 gold grids (Quantifoil, Großlöbichau, Germany) and plunge frozen in liquid ethane using an FEI Vitrobot Mark IV (Thermo Fisher Scientific) was used with 4°C, 100% humidity, 1 blot force, a wait time of ~5 s, and a 3 s blot time.

Grids were clipped and sent to Thermo Fisher Scientific RnD division in Eindhoven, The Netherlands, for data collection. Grids were loaded onto a Krios G4 microscope equipped with a Cold Field Emission gun and operated at 300 kV. Data were collected on a Falcon 4 detector mounted behind a Selectris X energy filter. The slit width of the energy filter was set to 10 eV; 7588 movie stacks containing 1251 frames were collected with EER (electron event representation) mode (*Guo et al., 2020*) of Falcon 4 detector at a magnification of 165,000 corresponding to a pixel size of 0.727 Å. Each movie stack was recorded with a total dose of 50 e$^-$/Å$^2$ on sample and a defocus range between 0.5 and 1.2 µm.

See *Supplementary file 1* for data collection statistics.

## Cryo-EM data processing

Motion correction and dose weighting were performed on all 7588 videos using RELION 4.0's implementation of MotionCor2 at 0.727 Å per pixel. Contrast transfer function (CTF) parameters were fit with CTFFIND-4.1. Template-free auto-picking of particles was performed with RELION 4.0's Laplacian-of-Gaussian filter on Video CTF fit to 5.0 Å or better, yielding an initial set of 2,379,507 particles. These particles were then extracted at a 288-pixel box size and transferred to cryoSPARC v.3.2 for 2D classification.

Iterative rounds of 2D classification resulted in a set of 31,126 particles which were then extracted in RELION 4.0 and their coordinates were used in the Topaz particle-picking pipeline (*Bepler et al., 2019*). Topaz training, picking, and extraction yielded 2,376,190 particles which were then subjected to one round of 2D classification in RELION 4.0 to remove obvious noise. The resulting 2,007,561 particles were extracted and then iteratively 2D classified in cryoSPARC v3.2, resulting in a set of 54,747 'good' particles.

Both the initial auto-picked particle set and subsequent Topaz particle set were lacking in good 2D classes of side views, so a subset of 13,698 particles from the best side view classes were extracted in RELION 4.0 and used to train and pick new particles in Topaz. As before, the resulting 2,186,648 particles were subjected to one round of 2D classification in RELION 4.0 then imported into cryoSPARC v3.2 for further 2D classification.

Good particles from the initial auto-picked particle set and both Topaz particle sets were pooled and duplicates within 100 Å were removed to yield 105,535 particles. These particles were extracted and imported into cryoSPARC v3.2 for three rounds of 2D classification to remove remaining junk. An ab initio reconstruction of the remaining 69,182 particles was performed to provide an initial volume and a subsequent non-uniform refinement (C2, 2 extra passes, 16 Å initial resolution) produced a map with a 4.0 Å overall resolution. This map was post-processed in RELION 4.0 and used for Bayesian particle polishing.

The resulting 'shiny' particles were imported back into cryoSPARC v3.2 for one additional round of 2D classification. The final 64,966 particles were used to generate a new ab initio and a subsequent non-uniform refinement (C2, 2 extra passes, 16 Å initial resolution, 1.5 adaptive window factor) yielded the final map at 3.5 Å nominal resolution.

## Modeling, refinement, and analysis

cryoSPARC sharpened cryo-EM maps were used to de novo model M using Coot (*Emsley et al., 2010*). The model was real space refined in Phenix (*Afonine et al., 2018*) and validated using Molprobity (*Williams et al., 2018*). Cavity and tunnel measurements were made with CASTp (*Tian et al., 2018*). Comparisons to the structure database were performed with DALI (*Holm, 2020*). Figures were prepared using ChimeraX (*Goddard et al., 2018*), Prism, Adobe Photoshop, and Adobe Illustrator software.

## Fluorescence size exclusion chromatography

Sf9 cells (~4 million) from the third day of infection were pelleted, frozen, and then thawed into extraction buffer (20 mM Tris pH 8, 150 mM KCl, all protease inhibitors used for protein purification, 1 mM EDTA, 1% DDM, 0.2% CHS). Extraction was performed at 4°C for 1 hr and lysate was then pelleted at 21,000 × *g* at 4°C for 1 hr to clear the supernatant. Supernatant was then run on a Superose 6 Increase column with fluorescence detection for GFP into 20 mM HEPES, pH 7.4, 150 mM KCl, 0.025% DDM, 0.005% CHS.

## Molecular dynamics

The initial MD system of M protein and lipid bilayer was built using CHARMM-GUI Membrane Builder (*Jo et al., 2008*; *Brooks et al., 2009*; *Lee et al., 2016*; *Jo et al., 2007*). A 20×20 nm lipid bilayer membrane was taken with a mixture of DOPE, POPS, and POPC lipids in a 2:1:1 mass ratio. A fully hydrated bilayer was built around the M protein, centering the transmembrane region close to the lipid bilayer center. To neutralize the system, a 0.15 M KCl salt concentration was used. The simulations were performed on GROMACS MD simulation package (*Abraham et al., 2015*) with the CHARMM36m force field (*Huang et al., 2017*). An initial minimization of the system was carried out following six-step protocols provided on CHARMM-GUI (*Jo et al., 2009*). A time step of 2fs was used

with periodic boundary conditions for the simulations. A simulation temperature of 310.15 K was maintained with a Nose-Hoover thermostat (*Hoover, 1985*; *Nosé, 1984*) and a coupling time constant of 1.0 ps in GROMACS. The pressure was set at 1 bar with a Berendsen barostat (*Berendsen et al., 1984*) during initial relaxation. For the production runs, the Parrinello-Rahman barostat was used semi-isotropically with the compressibility of $4.5 \times 10^{-5}$ and a coupling time constant of 5.0 ps (*Nosé and Klein, 1983*; *Parrinello and Rahman, 1981*). For the non-bonded interactions a switching function between 1.0 and 1.2 nm was used. The long-range electrostatics were computed using particle Mesh Ewald (*Darden et al., 1993*). The LINCS algorithm was used to constrain hydrogen bonds (*Hess et al., 1997*). We performed 1.6 µs production run for the system and used Frontera (TACC), and Midway2 (Research Computing Center at the University of Chicago) to run these simulations.

The RMSD of the protein and RMSF per residue (*Figure 5B and C*) were calculated using the GROMACS module. The center-of-mass distances between two residues (*Figure 5G and H*), number of Cα contacts between two monomers (*Figure 5D*), and angles between transmembrane and cytosolic regions (*Figure 5F*) were also calculated using the GROMACS package (*Abraham et al., 2015*). The analysis of the M pocket was performed using the HOLE program (*Smart et al., 1996*) implemented in MDAnalysis (*Michaud-Agrawal et al., 2011*; *Figure 5I and J*). In *Figure 5J*, each frame was taken at a 4 ns time step. The lipid distribution around the M protein was calculated using the MDAnalysis Python packages (*Michaud-Agrawal et al., 2011*; *Figure 5—figure supplement 1*). Visual Molecular Dynamics (VMD) (*Humphrey et al., 1996*) and PyMOL were used as visualization software.

## Acknowledgements

We thank Dan Toso, Jonathan Remis, and Paul Tobias for support collecting preliminary data at the Cal-Cryo EM facility. We thank Thermo Fisher Scientific for microscope access. We thank Diana Bautista and members in Brohawn and Bautista labs for feedback on the project. We thank Savitha Sridharan and Hillel Adesnik for the initial M DNA. SGB is a New York Stem Cell Foundation-Robertson Neuroscience Investigator. This work was funded in part by the New York Stem Cell Foundation, a Sloan Research Fellowship (to SGB), a Fast Grants Award from Emergent Ventures at the Mercatus Center, George Mason University (to SGB, Diana Bautista, and Hillel Adesnik at UC Berkeley), an NSF Graduate Research Fellowship (to KD), and an NSF RAPID grant CHE-2029092 (MD and GAV). Computer simulations were carried out on the Frontera supercomputer at the Texas Advanced Computer Center (TACC) as funded by the National Science Foundation (OAC-1818253), as well as on the Midway2 cluster at the Research Computing Center (RCC) of the University of Chicago.

## Additional information

### Competing interests

Abhay Kotecha: is an employee of Thermo Fisher Scientific. The other authors declare that no competing interests exist.

### Funding

| Funder | Grant reference number | Author |
|---|---|---|
| New York Stem Cell Foundation | R-N145 | Stephen G Brohawn |
| Fast Grants | | Stephen G Brohawn |
| National Science Foundation | | Kimberly A Dolan |
| National Science Foundation | CHE-2029092 | Mandira Dutta |
| National Science Foundation | OAC-1818253 | Gregory A Voth |
| Thermo Fisher Scientific | | Abhay Kotecha |

| Funder | Grant reference number | Author |
|--------|------------------------|--------|

The funders had no role in study design, data collection and interpretation, or the decision to submit the work for publication.

## Author contributions

Kimberly A Dolan, Conceived of the project, generated final constructs and performed expression, purification, reconstitution, cryo-EM sample preparation, and cryo-EM data processing, built and refined the atomic model, and wrote the manuscript with input from all authors; Mandira Dutta, Designed, performed, and analyzed molecular dynamics simulations; David M Kern, Conceived of the project and generated initial constructs and performed expression testing; Abhay Kotecha, Collected cryo-EM data; Gregory A Voth, Designed, performed, and analyzed molecular dynamics simulations, secured funding, and supervised research; Stephen G Brohawn, Conceived of the project, built and refined the atomic model, secured funding and supervised research, and wrote the manuscript with input from all authors

## Author ORCIDs

Kimberly A Dolan http://orcid.org/0000-0002-9335-7791
Gregory A Voth http://orcid.org/0000-0002-3267-6748
Stephen G Brohawn http://orcid.org/0000-0001-6768-3406

## Decision letter and Author response

Decision letter https://doi.org/10.7554/eLife.81702.sa1
Author response https://doi.org/10.7554/eLife.81702.sa2

## Additional files

### Supplementary files
• Supplementary file 1. Cryo-EM data collection, processing, refinement, and modeling data.
• MDAR checklist

### Data availability
All data and reagents associated with this study are publicly available. The final model is in the PDB under 8CTK, the final map is in the EMDB under EMD-26993, and micrographs (original and motion corrected) and final particle stack are deposited in EMPIAR under 11067.

The following datasets were generated:

| Author(s) | Year | Dataset title | Dataset URL | Database and Identifier |
|-----------|------|---------------|-------------|-------------------------|
| Dolan KA, Brohawn SG | 2022 | Cryo-EM structure of SARS-CoV-2 M protein in a lipid nanodisc | https://www.rcsb.org/structure/8CTK | RCSB Protein Data Bank, 8CTK |
| Dolan KA, Brohawn SG | 2022 | Cryo-EM structure of SARS-CoV-2 M protein in a lipid nanodisc | https://www.ebi.ac.uk/emdb/EMD-26993 | Electron Microscopy Data Bank, EMD-26993 |
| Dolan KA, Dutta M, Kern DM, Kotecha A, Voth GA, Brohawn SG | 2022 | Structure of SARS-CoV-2 M protein in lipid nanodiscs | https://www.ebi.ac.uk/empiar/EMPIAR-11067/ | Electron Microscopy Public Image Archive, EMPIAR-11067 |

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
