## [Editor Report]

The SARS-CoV2 M protein is an abundant protein in the viral envelope and is a potential target for vaccine and therapeutic development. M is one of only four structural proteins that are incorporated into mature SARS-CoV2 virions. This paper describes a single-particle cryo-electron microscopy structure of M in lipid nanodiscs. M forms a dimer similar to the previously characterized SARS-CoV2 ORF3a viroporin, which was proposed to function as an ion channel. Structural analysis and molecular dynamics simulations indicate that, unlike ORF3a, M functions as a structural scaffold protein. The highly charged surface of the M's cytosolic domain suggests how it might interact with other structural proteins and/or the charged RNA genome during virion packaging.

---

## [Decision Letter]

**Decision letter after peer review:**

Thank you for submitting your article "Structure of SARS-CoV-2 M protein in lipid nanodiscs" for consideration by *eLife*. Your article has been reviewed by 2 peer reviewers, and the evaluation has been overseen by a Reviewing Editor and Sara Sawyer as the Senior Editor. The reviewers have opted to remain anonymous.

Essential revisions:

The Reviewers are in agreement that the paper reports an important structure. Both reviewers highlight some questions that they would like the authors to address in a revision:

1) Please elaborate on the time scale of the MD simulation and how it is informative on protein function.

2) Please update the discussion on available functional data, in light of the new structure.

*Reviewer #1 (Recommendations for the authors):*

I have no strong comments or criticisms. The work is very well done, the writing is clear, and the figures are well designed and very clear. The work is overall relatively simple, with the structure being the main result and the supporting molecular dynamics contributing somewhat.

One point I would hope that the authors could clarify concerns the molecular dynamics simulations: These were apparently carried out to determine whether the M protein is flexible, as had been suggested by some prior (indirect) data. But, is 1.6 microseconds enough time to confidently say that the M protein is conformationally stable? I'm not familiar with the literature concerning MD to study conformational changes, but my naive sense is that one needs to do a very long simulation to see large-scale conformational changes. However, I'm not sure what "very long" means in this context. Some clarification, and perhaps a bit more explanation in the text, would be helpful.

I am hopeful that the authors are using their ability to purify this protein for further interesting experiments, which need not be included in this work but would nonetheless be very interesting for future studies. For example, direct interaction assays with the cytosolic domain of the S protein or the C-terminal domain of the N protein. Biochemical and even structural characterization of these interactions, should they prove to be specific enough, would be extremely exciting.

*Reviewer #2 (Recommendations for the authors):*

Dolan and colleagues report cryo-EM structure of the SARS-CoV-2 M protein. It is an impressive achievement in that the authors obtained ~3.5 Å resolution structure of a small 50 kDa membrane protein. The structural data and analyses were clearly presented in this manuscript. Molecular dynamics simulations supported the rather rigid structural scaffold of the M protein. One concern is that the study did not include any functional data of M, thus providing no mechanistic insight into its role in viral assembly. Otherwise, the manuscript is well written and suitable for publication.

---

## [Author Response]

Essential revisions:The Reviewers are in agreement that the paper reports an important structure. Both reviewers highlight some questions that they would like the authors to address in a revision:1) Please elaborate on the time scale of the MD simulation and how it is informative on protein function.

We have increased the duration of the MD simulation to 2 us and elaborated on the significance of the time scale in the revised manuscript. Major conformational changes related to protein function typically take place on a timescale ranging from a microsecond to seconds^64-66^. Numerous studies, including our previous work, have reported microsecond simulations can capture large conformational changes of membrane proteins in a lipid bilayer^67-70^. We therefore expected to capture a subset of possible conformational dynamics in our simulation that correspond to more frequently accessed states. We do not observe significant changes in M protein structure in the simulation, suggesting M adopts a relatively stable structure with minimal dynamic rearrangement on time scale, at physiological temperature, and in the absence of additional binding proteins. This is consistent with the absence of other highly populated conformations in our cryo-EM data. We cannot exclude the possibility that M undergoes rare large conformational changes under these conditions that are not captured in the simulation.

2) Please update the discussion on available functional data, in light of the new structure.

We have updated the discussion in light of the new structural and functional data from a complementary paper published during review of this manuscript (Zhang et al., 2022).